# MULTI-VIEW DISENTANGLED REPRESENTATION

## ABSTRACT

Learning effective representations for data with multiple views is crucial in machine learning and pattern recognition. Recent great efforts have focused on learning unified or latent representations to integrate information from different views for specific tasks. These approaches generally assume simple or implicit relationships between different views and as a result are not able to accurately and explicitly depict the correlations among these views. To address this, we firstly propose the definition and conditions for unsupervised multi-view disentanglement providing general instructions for disentangling representations between different views. Furthermore, a novel objective function is derived to explicitly disentangle the multi-view data into a shared part across different views and a (private) exclusive part within each view. The explicit guaranteed disentanglement is of great potential for downstream tasks. Experiments on a variety of multi-modal datasets demonstrate that our objective can effectively disentangle information from different views while satisfying the disentangling conditions.

## 1 INTRODUCTION

Multi-view representation learning (MRL) involves learning representations by effectively leveraging information from different perspectives. The representations produced by MRL are effective when correlations across different views are accurately modeled and thus properly exploited for downstream tasks. One representative algorithm, Canonical Components Analysis (CCA) (Hotelling, 1992), aims to maximize linear correlations between two views under the assumption that factors from different views are highly correlated. Under a similar assumption, the extended versions of CCA, including kernelized CCA (Akaho, 2006) and Deep CCA (Andrew et al., 2013), explore more general correlations. There are also several methods (Cao et al., 2015; Sublime et al., 2017) that maximize the independence between different views to enhance the complementarity. Going beyond the simple assumptions above, the latent representation encodes different views with a degradation process implicitly exploiting both consistency and complementarity (Zhang et al., 2019).

These existing MRL algorithms are effective, however, the assumed correlations between different views are usually simple thus cannot accurately model or explicitly disentangle complex real-world correlations, which hinders the further improvement and interpretability. Although there are a few heuristic algorithms (Tsai et al., 2019; Hu et al., 2017) that explicitly decompose the multi-view representation into shared and view-specific parts, they are especially designed for supervised learning tasks without any disentangled representation guarantee and fall short in formally defining the relationships between different parts. To address this issue, we propose to unsupervisedly disentangle the original data from different views into shared representation across different views and exclusive (private) part within each view, which explicitly depicts the correlations and thus not only enhances the performance of existing tasks but could also inspire potential applications. Specifically, we firstly provide a definition for the multi-view disentangled representation by introducing the sufficient and necessary conditions for guaranteeing the disentanglement of different views. According to these conditions, an information-theory-based algorithm is proposed to accurately disentangle different views. To summarize, the main contributions of our work are as follows:

- To the best of our knowledge, this is the first work to formally study multi-view disentangled representation with strict conditions, which might serve as the foundations of the future research on this problem.
- Based on our definition, we propose a multi-view disentangling model, in which information-theory-based multi-view disentangling can accurately decompose the information into shared

(a) Original data | (b) ①+②+③+④ | (c) ②+③+④ | (d) ①+③+④ | (e) ①+②+④ | (f) ①+②+③

Figure 1: Illustration of multi-view disentangled representation. (a): The red and white graphics indicate the shared information between different views, and the (private) exclusive information within each view, respectively. (b): The exact disentangled representation can be achieved - the shared (gray area) and exclusive (white area) components are separated, when the four conditions in definition 2.1 are satisfied. (c)(d)(e)(f): The exact disentangled representation cannot be guaranteed when any condition is violated. Intuitively, the proposed four conditions are necessary and sufficient conditions since any change of (b) will violate the definition.

- representation across different views and exclusive representation within each view. The explicit decomposition enhances the performance of multi-view analysis tasks and could also inspire new potential applications.
- Different from the single-view unsupervised disentangled representation learning (Locatello et al., 2019), we provide a new paradigm for unsupervised disentangled representation learning from a fresh perspective - disentangling factors between different views instead of each single view.
- Extensive experiments on a range of applications verify that the proposed information-theory-based multi-view disentangling algorithm can accurately disentangle data from multiple views into expected shared and exclusive representations.

## 2 MULTI-VIEW DISENTANGLED REPRESENTATION

Existing multi-view representation learning methods (Wu & Goodman, 2018; Zhang et al., 2019) can obtain a common representation for multi-view data, however, the correlations between different views are not explicitly expressed. The supervised algorithms (Hu et al., 2017; Tan et al., 2019) can decompose multiple views into a common part and private parts, but there is no disentangling guarantee. Therefore, we propose a multi-view disentanglement algorithm that can explicitly separate the shared and exclusive information in unsupervised settings. Formally, we first propose a definition of a multi-view disentangled representation by introducing four criteria, which are considered as sufficient and necessary conditions of disentangling multiple views. The definition is as follows:

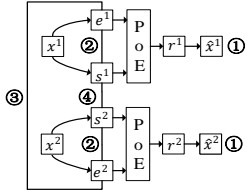

Figure 2: Illustration of our model, which corresponds to the objective in Eq. 1 and the conditions in definition 2.1. Refer to the text for PoE (Product of Expert) in 2.1.

**Definition 2.1** *(**Multi-View Disentangled Representation**) Given a sample with two views, i.e., $\mathcal{X} = \{x^i\}_{i=1}^2$, the representation $S_{dis} = \{s^i, e^i\}_{i=1}^2$ is a multi-view disentangled representation if the following conditions are satisfied:*

- ***Completeness:*** ① *The shared representation $s^i$ and exclusive representation $e^i$ should jointly contain all information of the original representation $x^i$;*
- ***Exclusivity:*** ② *There is no shared information between common representation $s^i$ and exclusive representation $e^i$, which ensures the exclusivity within each view (intra-View). ③ There is no shared information between $e^i$ and $e^j$, which ensures the exclusivity between private information of different views (inter-View).*
- ***Commonality:*** ④ *The common representations $s^i$ and $s^j$ should contain the same information. Equipped with the exclusivity constraints, the common representations are guaranteed to not only be the same but also contain maximized common information.*

The necessity for each criterion is illustrated in Fig. 1 (satisfaction of all the four conditions produces exact disentanglement, and violation of any condition may result in an unexpected disentangled representation). Note that, existing (single-view) unsupervised disentanglement focuses on learning

a representation to identify explanatory factors of variation, which has been proved fundamentally impossible (Locatello et al., 2019). *The goal of the proposed multi-view disentanglement is to disentangle multiple views into the shared and exclusive parts which can be well guaranteed as illustrated in definition 2.1 and Fig. 1.*

Mutual information has been widely used in representation learning (Hjelm et al., 2019; Belghazi et al., 2018). In probability theory and information theory, the mutual information of two random variables quantifies the "amount of information" obtained about one random variable when observing the other one, which is well-suited for measuring the amount of shared information between two different views. To approach the disentangling goal, according to conditions ①~④, the general form of the object function is naturally induced as:

$$\max \sum_{i,j=1}^{2} \big[ \underbrace{\mathcal{I}(x^i; e^i, s^i)}_{①} - \underbrace{\mathcal{I}(e^i; s^i)}_{②} \big] - \sum_{i \neq j} \underbrace{\mathcal{I}(e^i; e^j)}_{③} + \sum_{i \neq j} \underbrace{\mathcal{I}(s^i; s^j)}_{④}, \tag{1}$$

where $\mathcal{I}(\cdot; \cdot)$ denotes the mutual information. We provide an implementation in Fig. 2 and, in the following subsections, we will describe this implementation in detail.

## 2.1 CONDITION ①: INFORMATION PRESERVATION FOR THE SHARED AND EXCLUSIVE REPRESENTATIONS

• **How to maximize $\mathcal{I}(x; e, s)$?**

For simplicity, $x$, $s$, $e$ and $x^i$, $s^i$, $e^i$ are denoted with the same meanings and used alternately, where the former and latter are used for intra-view and inter-view cases, respectively. To preserve the information from the original data in the shared and exclusive representations, the mutual information $\mathcal{I}(x; e, s)$ should be maximized. There are different ways to implement the maximization of $\mathcal{I}(x; e, s)$ based on the following assumptions.

**Assumption 2.1** *The shared representation $s$ and exclusive representation $e$ are simultaneously independent and conditionally independent:*

$$p(s, e) = p(s)p(e), \ p(s, e|x) = p(s|x)p(e|x). \tag{2}$$

Firstly, we expand $\mathcal{I}(x; e, s)$ to obtain the following equation (more details are shown in supplement C):

$$\mathcal{I}(x; e, s) = \int \int \int p(x)p(e, s|x) \log \frac{p(e, s|x)}{p(e, s)} de \, ds \, dx.$$

Then, under Assumption 2.1, the following equation is derived (more details are shown in supplement C):

$$\mathcal{I}(x; e, s) = \mathcal{I}(x; e) + \mathcal{I}(x; s). \tag{3}$$

According to the above equation, it seems that we can maximize $\mathcal{I}(x; e) + \mathcal{I}(x; s)$ to maximize $\mathcal{I}(x; e, s)$, which involves making $s$ and $e$ contain as much information from $x$ as possible (ideally, it will produce $e$ and $s$ to meet $\mathcal{I}(x; e) = \mathcal{I}(x; s) = H(x)$, where $H(x)$ is the entropy of $x$). This actually leads to a strong correlation between $s$ and $e$, which is in conflict with the independence Assumption 2.1 about $s$ and $e$. In other words, it is difficult to balance the completeness (condition ①) and intra-view exclusivity (condition ②) (see experimental results in supplement B.4).

Fortunately, there is an alternative strategy which avoids the difficulty in balancing the completeness and intra-view exclusivity. Specifically, we introduce a latent representation $r$ generated by two independent distributions with respect to $s$ and $e$ under a mild assumption:

**Assumption 2.2** *(Relationship between $s, e$ and $r$):*

$$p(s, e, x) = p(r, x). \tag{4}$$

In our formulation, we define $r = f(s, e)$, where $r$ is derived from $s$ and $e$ with the underlying function $f(\cdot)$ and satisfies $p(r, x) = p(s, e, x)$. Eq. 4 is a mild assumption, for example the invertibility of mapping $r = f(s, e)$ ensuring a sufficient condition which can be easily verified. Note that $r = [s, e]$

is one special case and will be discussed later. Based on Assumption 2.2, we can get (more details are shown in supplement C):

$$p(r) = p(s, e), p(r|x) = p(s, e|x). \tag{5}$$

Then, we can induce the following result (more details are shown in supplement C):

$$\mathcal{I}(x; e, s) = \mathcal{I}(x; r). \tag{6}$$

This result indicates that the maximization of $\mathcal{I}(x; e, s)$ can be achieved by maximizing the mutual information of agency $r$ and $x$. In this way, the independence of $e$ and $s$ is well preserved and the previous conflict is dispelled. Next, we will explain how to encode the information of $x$ into independent representations $s$ and $e$ by introducing the agency $r$.

• **How to obtain independent representations $e$ and $s$ by maximizing $\mathcal{I}(x; r)$ ?**

First, we consider encoding the observed data $x$ into a latent representation $r$ by maximizing the mutual information between $x$ and $r$. Considering robustness and effectiveness (Alemi et al., 2018), we can maximize the mutual information between $r$ and $x$ through Variational Autoencoders (VAEs) (Kingma & Welling, 2014). Accordingly, we have the following objective function:

$$\min_{q_r, d} \mathbb{E}_{x \sim p(x)} \Big[ - \mathbb{E}_{r \sim q_r(r|x)} \big[ log(d(x|r)) \big] + \mathbb{E}_{r \sim q_r(r|x)} \log \frac{q_r(r|x)}{p(r)} \Big], \tag{7}$$

where $d(x|r)$ (the "decoder") is a variational approximation to $p(x|r)$, and $q_r(r|x)$ (the "encoder") is a variational approximation to $p(r|x)$, which converts the observed data $x$ into the latent representation $r$.

Second, we consider how to obtain independent representations $e$ and $s$ by modeling $q_r(r|x)$. For this goal, the relationships between $s$, $e$ and $r$ should be jointly modeled. As shown in Eq. 5, we obtain $p(r|x) = p(s, e|x)$. Under Assumption 2.1, Eq. 5 can be rewritten as $p(r|x) = p(s|x)p(e|x)$, which implies that $q_r(r|x)$ can be considered as the product of $p(s|x)$ and $p(e|x)$. Furthermore, we introduce PoE (product-of-experts) (Hinton, 2002; Wu & Goodman, 2018) to model the product of $q_s(s|x)$ and $q_e(e|x)$, where the variational networks $q_s(s|x)$ and $q_e(e|x)$ are designed to approximate $p(s|x)$ and $p(e|x)$. It is worth noting that the key difference from MVAE (Multimodal Variational Autoencoder) (Wu & Goodman, 2018) is that our model obtains the latent representation $r$ from two independent components within each single view, while MVAE achieves the unified representation of all views by assuming independence of representations of different views. Under the assumption that the true posteriors for individual factors $p(s|x)$ and $p(e|x)$ are contained in the family of their variational counterparts $q_s(s|x)$ and $q_e(e|x)$, we have $q_r(r|x) = q_s(s|x)q_e(e|x)$. With Gaussian distributions, we can obtain the closed-form solution for the product of two distributions: $\mu_r = \frac{\mu_s \sigma_s^2 + \mu_e \sigma_e^2}{\sigma_s^2 + \sigma_e^2}$, $\sigma_r^2 = \frac{\sigma_s^2 \sigma_e^2}{\sigma_s^2 + \sigma_e^2}$. Therefore, the independent representations between $e$ and $s$ are well preserved by modeling $q_r(r|x)$.

Accordingly, with the results $q_r(r|x) = q_s(s|x)q_e(e|x)$ and $p(r) = p(s)p(e)$, the objective in Eq. 7 is rewritten as:

$$\min_{q_s, q_e, d} \mathbb{E}_{x \sim p(x)} \Big[ - \mathbb{E}_{r \sim q_s(s|x)q_e(e|x)} \big[ log(d(x|r)) \big]$$
$$+ \mathbb{E}_{s \sim q_s(s|x)} \log \frac{q_s(s|x)}{p(s)} + \mathbb{E}_{e \sim q_e(e|x)} \log \frac{q_e(e|x)}{p(e)} \Big],$$

where $p(s)$ and $p(e)$ are set to Gaussian distributions, which in turn forces $q_s(s|x)$ and $q_e(e|x)$ to be closer to a Gaussian distribution, allowing us to find the product of the two distributions. The above objective is actually the ELBO (evidence lower bound) (Kingma & Welling, 2014) with the first term being the reconstruction loss, and the second and third terms being the KL divergence.

The proposed variant of VAE inherits two advantages from VAE and PoE, respectively. The first is that we can obtain approximate distributions of $s$ and $e$ given $x$ to preserve the independence. The second is that the proposed model still works even when there is a missing case for $e$ or $s$ in the testing. This means that we can use only $s$ or $e$ as input to the decoder to reconstruct $x$ (shown in the experimental section), which is quite different from the concatenation of $e$ and $s$ or other forms that require $e$ and $s$ simultaneously to obtain $r$. In addition, the way of concatenating $s$ and $e$ does not well exploit the independent prior of $s$ and $e$.

## 2.2 CONDITIONS ②-③: EXCLUSIVITY

To fulfill conditions ② and ③, we minimize the mutual information between two variables by enhancing the independence. There are different strategies to promote the independence between variables, which are endowed with different properties. Specifically, the straightforward way is to promote the independence by minimizing the linear correlation. Accordingly, we have the following loss function:

$$\min_{q_e^i, q_e^j} \frac{e^i e^{jT}}{\|e^i\| \|e^j\|}, \tag{8}$$

for condition ②, and a similar objective could be induced for condition ③. Although simple and effective, the linearity property may not be powerful enough to handle complex real-world complex correlations. Therefore, we also propose an alternative strategy for general correlation cases in the supplementary material A.1.

## 2.3 CONDITION ④: ALIGNMENT OF THE SHARED REPRESENTATION FROM DIFFERENT VIEWS

For condition ④, we ensure the commonality between $s^i$ and $s^j$ by maximizing the mutual information as follows:

$$\mathcal{I}(s^i; s^j) = \int \int p(s^i, s^j) \log \frac{p(s^i, s^j)}{p(s^i)p(s^j)} ds^i ds^j.$$

It is difficult to calculate the mutual information, since the true distribution is usually unknown. Based on the scalable and flexible MINE (Belghazi et al., 2018), we introduce two different strategies for maximizing the mutual information between the shared representations $s^i \sim q_s^i(s^i|x^i)$ and $s^j \sim q_s^i(s^j|x^j)$ from different perspectives.

MINE can estimate mutual information of two variables by training a classifier to distinguish whether the samples come from the joint distribution $\mathbb{J}$ or the product of marginals $\mathbb{M}$. MINE actually aims to optimize the tractable lower bound to estimate mutual information based on the Donsker-Varadhan representation (Donsker & Varadhan, 1983) of the KL-divergence, with the following form:

$$\mathcal{I}(s^i, s^j) \geq \mathbb{E}_{\mathbb{J}} \left[ T_\theta(s^i, s^j) \right] - \log \left( \mathbb{E}_{\mathbb{M}} \left[ e^{T_\theta(s^i, s^j)} \right] \right),$$

where $T_\theta$ is a discriminator function modeled by a neural network with parameters $\theta$. $\mathbb{J}$ and $\mathbb{M}$ are the joint and product of marginals, respectively. We can maximize the mutual information of $s^i$ and $s^j$ by maximizing the lower bound.

Although the KL-based MI is effective for some tasks, it tends to overemphasize the similarity between samples and thus cannot thoroughly explore the underlying similarity between different distributions. To address this issue, we could replace the KL divergence with JS divergence (Hjelm et al., 2019), which can focus on the similarity in terms of different distributions instead of samples. Accordingly, we maximize the mutual information of $s^i$ and $s^j$ by the following form:

$$\max_{q_s^i, q_s^j, T_\theta} \mathbb{E}_{\mathbb{J}} \left[ - \operatorname{sp} \left( -T_\theta(s^i, s^j) \right) \right] - \mathbb{E}_{\mathbb{M}} \left[ \operatorname{sp} \left( T_\theta(s^i, s'^j) \right) \right],$$

where $s^i, s^j$ corresponds to one sample with two views, i.e., the $i$th and $j$th views, respectively. $s'^j$ corresponds to another sample from the $j$th view, and $\operatorname{sp}(z) = \log(1 + e^z)$ is the softplus function. Specifically, the inner product is employed in the classifier, i.e., $T_\theta(a, b) = a^T b$. We have discussed these two methods in the supplementary material A.2.

## 3 RELATED WORK

The disentanglement of representations aims to depict an object through independent factors in order to provide a more reliable and interpretable representation (Bengio et al., 2013). Most unsupervised disentangled representation learning methods (Chen et al., 2018; Esmaeili et al., 2019; Kumar et al., 2018) are based on Variational Autoencoders (VAEs) (Kingma & Welling, 2014). Existing VAE-based methods basically increase the independence between different factors of the representation. On the basis of VAEs, $\beta$-VAE (Higgins et al., 2017) implicitly obtains promising disentanglement

performance by increasing $\beta$ in ELBO to constrain the capacity of the latent space. Different from $\beta$-VAE, several methods (Chen et al., 2018; Esmaeili et al., 2019) increase the independence between different factors by minimizing a total correlation (TC) loss. DIP (Disentangled Inferred Prior) (Kumar et al., 2018) encourages disentanglement by introducing a disentangled prior to constrain the disentangled representation. However, there are theoretical problems in unsupervised disentanglement learning (Locatello et al., 2019). Thus there are also several semi-supervised methods (Kingma et al., 2014; Narayanaswamy et al., 2017; Bouchacourt et al., 2018) for disentanglement representation learning which have access to partial real factors of the data. There are also various real-world applications using disentangled representations (Gonzalez-Garcia et al., 2018; Liu et al., 2018).

Multi-view representation learning aims to jointly utilize information from multiple views for better performance. To jointly learn a unified representation between multiple views, CCA-based (Hotelling, 1992; Akaho, 2006; Andrew et al., 2013; Wang et al., 2015) algorithms maximize the correlation between different views to extract shared information. KCCA (Akaho, 2006) and DCCA (Andrew et al., 2013) extend the traditional CCA using kernel and deep neural networks, respectively. DCCAE (Wang et al., 2015) jointly considers the reconstruction of each single view and the correlation across different views. To jointly encode the shared and view-specific information, some latent-representation-based models (Zhang et al., 2019) have been proposed. There are also models (Wu et al., 2019; Suzuki et al., 2016; Vedantam et al., 2018) that employ a VAE to learn a unified multi-modal representation.

## 4 Experiments

**Experimental Settings.** We conduct quite comprehensive experiments to evaluate the disentangled representation. Specifically, we investigate the disentanglement quantitively by conducting clustering and classification (section 4.1), and provide visualization results to intuitively evaluate the disentanglement (section 4.2 and section B.5). Furthermore, we conducted ablation experiments (section B.3) and present application experiments based on the disentangled representation (section B.6). Due to the space limitation, some experiments are detailed in the supplementary material.

**Datasets:** Similar to the work (Gonzalez-Garcia et al., 2018), we construct the dataset **MNIST-CBCD**, comprising MNIST-CB (MNIST with Colored Background) and MNIST-CD (MNIST with Colored Digit) as two views, by randomly modifying the color of digits and background of digit images from the dataset **MNIST**. *Intuitively, the shape of a digit corresponds to the shared information, while the colors of background and digit correspond to the private information within each view.* The same strategy is applied to **FashionMNIST**. We also conduct experiments on the face image dataset **CelebA** (Liu et al., 2015), which is a large-scale face-attributes dataset with more than 200K celebrity images, each of which is with 40 attribute annotations. The image and attribute domains are considered as two different views. We select the 18 most significant attributes as the attribute vector (Perarnau et al., 2016). *For these two views, the shared information are attribute-related information (e.g., viewpoint, hair color, w/ or w/o glasses etc), and the exclusive representation is non-attribute information.*

To verify the disentanglement, we compare our algorithm with: (1) Raw-data (Raw), which reshapes images directly into vectors as representations; (2) Variational autoencoders (VAE (Kingma & Welling, 2014)), which uses VAE to extract features from the data of each view; (3) CCA-based methods (CCA (Hotelling, 1992), KCCA (Akaho, 2006), DCCA (Andrew et al., 2013) and DCCAE (Wang et al., 2015)), which obtain the common representation by maximizing the correlation between different views. (4) Multimodal variational autoencoders (MVAE (Wu et al., 2019)), which can learn a common representation of two views. For Raw-data and VAE, we report the clustering results using the representations obtained by view-1, view-2, and representation by concatenating view-1 and view-2. In our method, we use $s^1$, $s^2$ and the concatenated representation for clustering/classification.

### 4.1 Quantitative Analysis

To evaluate the disentanglement of our algorithm, we conduct clustering and classification based on the representations, respectively. For simplicity, we employ k-means as the clustering algorithm, since k-means is based on the Euclidean distance, which makes it more objective in measuring the quality of representations. KNN (K-Nearest Neighbour) and linear SVM are employed to conduct

| | | Raw | | | VAE | | | CCA-based | | | MVAE | Ours | | |
|---|---|---|---|---|---|---|---|---|---|---|---|---|---|---|---|
| | | view1 | view2 | Concat | view1 | view2 | Concat | CCA | KCCA | DCCA | DCCAE | | view1 | view2 | Concat |
| M | ACC | 11.7 | 26.8 | 12.2 | 35.4 | 38.5 | 56.2 | 46.9 | 48.2 | 40.2 | 47.4 | 50.4 | **62.0**[3] | **62.1**[2] | **62.4**[1] |
| | NMI | 0.30 | 18.2 | 0.50 | 25.0 | 30.6 | 50.2 | 54.2 | 52.9 | 51.3 | 51.9 | 41.9 | **54.8**[3] | **56.4**[2] | **57.0**[1] |
| F | ACC | 15.1 | 36.8 | 28.7 | 42.5 | 45.6 | 52.0 | 48.6 | 46.2 | 42.5 | 46.4 | 51.8 | **56.1**[2] | **54.9**[3] | **56.3**[1] |
| | NMI | 3.90 | 32.3 | 24.5 | 41.1 | 45.1 | 55.1 | 55.1 | 56.7 | 52.8 | 54.0 | 54.1 | **58.2**[3] | **58.5**[2] | **59.4**[1] |

Table 1: Comparison on the clustering task. 'M' and 'F' indicate MNIST-CBCD and FMNIST-CBCD, respectively. The top three results are in bold and marked with superscript.

| | | Raw | | | VAE | | | CCA-based | | | MVAE | Ours | | |
|---|---|---|---|---|---|---|---|---|---|---|---|---|---|---|---|
| | | view1 | view2 | Concat | view1 | view2 | Concat | CCA | KCCA | DCCA | DCCAE | | view1 | view2 | Concat |
| KNN | M | 77.8 | **90.6**[2] | 77.9 | 78.7 | 89.3 | 79.2 | 83.0 | 55.5 | 52.9 | 59.5 | 87.2 | 88.1 | **89.9**[3] | **91.2**[1] |
| | F | 70.7 | 75.2 | 67.8 | 74.8 | **78.7**[1] | 74.9 | 64.1 | 53.3 | 52.8 | 56.8 | 73.2 | 74.2 | **78.1**[2] | **77.5**[3] |
| LSVM | M | 82.3 | 90.4 | **91.5**[3] | 85.6 | 91.0 | 91.1 | 83.5 | 61.8 | 51.6 | 64.9 | **92.4**[1] | 89.2 | 90.5 | **92.2**[2] |
| | F | 59.7 | 78.1 | 79.6 | 77.9 | 81.3 | **81.4**[3] | 65.1 | 57.4 | 47.1 | 59.0 | 79.8 | 79.4 | **81.9**[1] | **81.9**[1] |

Table 2: Comparison on the classification task. 'KNN' and 'LSVM' indicate the K-Nearest Neighbor and linear SVM respectively. 'M' and 'F' indicate MNIST-CBCD and FMNIST-CBCD, respectively

| | MNIST-CBCD | FMNIST-CBCD |
|---|---|---|
| View-1 ($e^1$) | **69.89$\pm$8.47** | **70.80$\pm$5.99** |
| Raw View-1 | 66.39$\pm$10.79 | 57.65$\pm$6.94 |
| View-2 ($e^2$) | **54.61$\pm$3.30** | **58.12$\pm$4.63** |
| Raw View-2 | 53.19$\pm$5.90 | 51.92$\pm$4.80 |

| | MVAE | Ours |
|---|---|---|
| top 1 | 64.12 | **95.15** |
| top 5 | 63.24 | **94.57** |
| top 10 | 62.85 | **94.16** |
| top 100 | 61.53 | **91.96** |

Table 3: Clustering with exclusive representation.  Table 4: Cross-modal retrieval.

classification experiments based on the shared representations. All experiments are run 20 times and the means are reported in terms of accuracy (refer to the supplement for standard deviations).

From the quantitative results in Tables 1 and 2, the following observations are drawn: (1) directly using the raw features for clustering/classification is not promising, as the digital and color information are mixed. Moreover, since the background region is much larger than the area of the digit, the accuracy of using view-1 is relatively low on MNIST; (2) compared with the raw features, the shared information extracted by our model is competitive due to the clear semantic information; (3) by extracting the shared (digit) information explicitly, our model obtains much better results.

Furthermore, we evaluate the exclusive representations on clustering. For the MNIST, the colors of background (MNIST-CB: MNIST with Colored Background) or digits (MNIST-CD: MNIST with Colored Digit) are considered as class labels. According to Table 3, our algorithm obtains more promising clustering performance with the exclusive representation compared with the raw data, while existing algorithms cannot obtain exclusive representation explicitly. The performance improvement of view-2 (on MNIST-CBCD) is not so substantial. The possible reason is that exclusive information (the color of digit) from the images is not so significant due to small area ratio of digits, which increases the difficulty of disentanglement.

We verify our disentangled representation with cross-modal retrieval on CelebA (Liu et al., 2015). Specifically, after training the disentangling networks, we can obtain the shared representations from the image and attribute views, respectively. Therefore, the attribute vector can be used to retrieve the related face images (attribute-specific cross-modal retrieval). The quantitative results are reported in Table 4, and examples are in Fig. 9(a) (in the supplement). Given the specific attributes represented as vector $l_n$, we can obtain attribute vector $\hat{l}_{nk}$ for the $k^{th}$ most similar retrieved image, which is associated with $D$ attributes (the value of each one is 0 or 1). Accordingly, for the top $K$ retrieved images, we have $accuracy = \frac{\sum_{n=1}^{N}\sum_{k=1}^{K}\sum_{d=1}^{D}\delta(l_{nkd},\hat{l}_{nkd})}{N \times K \times D}$, where $\delta(a,b) = 1$ when $a = b$, otherwise $\delta(a,b) = 0$. According to the results in Table 4, the performances of our model are much higher than those of MVAE due to the promising disentanglement.

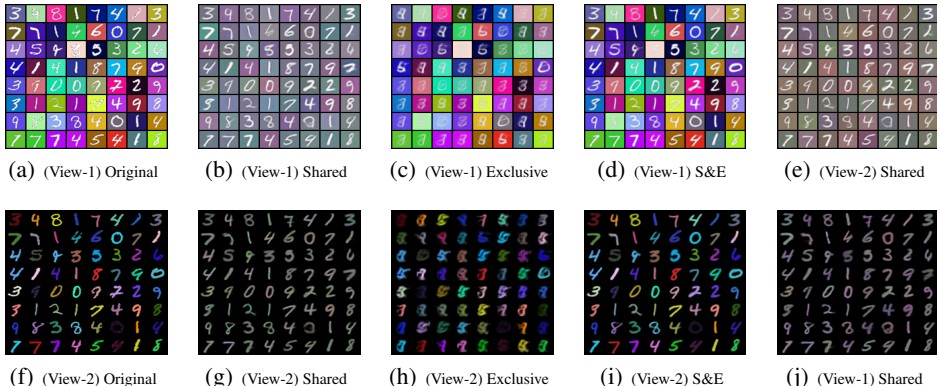

(a) (View-1) Original  (b) (View-1) Shared  (c) (View-1) Exclusive  (d) (View-1) S&E  (e) (View-2) Shared

(f) (View-2) Original  (g) (View-2) Shared  (h) (View-2) Exclusive  (i) (View-2) S&E  (j) (View-1) Shared

Figure 3: Visualization of reconstruction with shared and exclusive representations. The top and bottom rows correspond to the reconstruction results from the decoders of view-1 and view-2, respectively. 'Shared', 'Exclusive' and 'S&E' indicate shared, exclusive, and the combination of shared and exclusive representations, respectively. 'View-1' and 'View-2' in the parentheses indicate the view where these representations come from. Note that, the images in (e) ((j)) are reconstructed results using decoder of view-1 (view-2) using the share representation from view-2 (view-1).

## 4.2 QUALITATIVE ANALYSIS

We further intuitively demonstrate that our model can well fulfill the four conditions in definition 2.1 with visualization. On MINIST-CBCD, we train the model using the training set and randomly select 64 images in the test set for visual analysis. The disentangled shared and exclusive representations are used as input to decoders corresponding to different views to reconstruct the original data with different combinations. For example, we can input the shared representation extracted from view-1 into the decoder corresponding to view-2 to obtain the reconstructed images from view-2. The visualization of reconstruction results are shown in Fig 3.

From Fig. 3, we have the following observations, which are consistent with the definition of multi-view disentanglement: (1) By combining the shared and exclusive information, the original image can be fully reconstructed ((d) and (i)), satisfying condition ① (completeness); (2) The shared and exclusive representations contain different information. With the shared representation, we can reconstruct images ((b) and (g)) with clear digit shapes rather than color information as in the original images. In contrast, with the exclusive representations, we can reconstruct the color information ((c) and (h)) of the original images rather than the digit shapes. This verifies that the condition ② (intra-view exclusivity) is satisfied. (3) The exclusive representations from different views contain different information. Specifically, the exclusive representation (c) from view-1 contains the information of background color, while the exclusive representation (h) from view-2 contains information of the digit color. This verifies that our model satisfies condition ③ (inter-view exclusivity). (4) The shared representations ((b), (g), (e) and (j)) from different views contain (almost) the same information, i.e., condition ④ (commonality). We verify this by reconstructing digit shapes in view-2 using the shared representations from view-1 and vice versa. Similar experiments are done on CelebA (section B.5).

## 5 CONCLUSION

In this work, we proposed a formal definition for disentangling multi-view data, and based on this developed a principled algorithm which focuses on automatically disentangling multi-view data into shared and exclusive representations without supervision. Extensive experiments validate that the proposed algorithm can promote subsequent analysis tasks (e.g., clustering/classification/retrieval). We consistently validated that the proposed algorithm can provide promising disentanglement and thus is quite effective and flexible in analyzing and manipulating multi-view data. We will focus on the semi-supervised setting to improve the discriminative ability in the future.

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

## Supplemental Materials: Multi-View Disentangled Representation

## A  SUPPLEMENTAL MATERIAL FOR METHODS

### A.1  SUPPLEMENTAL MATERIAL FOR CONDITIONS ②-③

Inspired by Choi et al. (2018), we introduce a classifier to distinguish these (independent) representations generated by the encoders. The loss function of the classification can be defined as:

$$\min_{q,C} \mathbb{E}_R \big[ - \int p(z) \log C(z|R) dz \big], \tag{9}$$

where $C$ is a classifier that distinguishes the representation from different sources (independent representations), $R$ is from a representation set with different sources, e.g., private representation $e^i$ and $e^j$ from different views, and $q$ corresponds to the encoder of different views. $z$ is a label which indicates the source of $R$.

Generally, it is difficult to strictly guarantee the independence; however, the two different strategies can promote the independence between different representations (generated by the encoders) to a certain extent. We implement both strategies and similar results are observed in practice.

### A.2  DISCUSSION OF CONDITION ④

Both KL- and JS-based estimators can maximize the mutual information. However, due to the different properties of the KL and JS divergence, the two estimators are suitable for different scenarios. Since the JS divergence is bounded, in theory, it prevents the estimator from overemphasizing the similarity of two representations for the same sample (even if they are exactly the same, it will not obviously reduce the loss). This prevents the encoder from paying too much attention to generating the exact $s^i$ coordination with $s^j$ instead of the overall objective function. In contrast, since the estimator based on the KL divergence is unbounded, $s^i$ and $s^j$ are forced to be as similar as possible. Although this is not appropriate for most tasks, it helps us to observe whether $s^i$ and $s^j$ intuitively have high mutual information. For example, we can replace $s^i$ and $s^j$ with each other to see if they can accomplish the same task (which is demonstrated in the experimental part).

## B  SUPPLEMENTAL EXPERIMENTS

### B.1  NETWORK ARCHITECTURES

For MNIST and FashionMNIST, two convolutional layers and two fully connected layers are used for the encoder, while we employ two fully connected layers and two deconvolution layers for the decoder. For the face image dataset CelebA, we use four convolutional layers and two fully connected layers to build the encoder for handling the image view, while the decoder is built using two fully connected layers and four deconvolutional layers. For the attribute vector view, three fully connected layers are used to construct both the encoder and decoder. The batch normalization Ioffe & Szegedy (2015) and Swish activation functions Ramachandran et al. (2017) are used between the convolutional layers.

### B.2  DETAILED EXPERIMENTAL RESULTS OF QUANTITATIVE EXPERIMENTS

Due to space limitations, we only report the means of clustering and classification experiments in the text, and here we add their standard deviations in Table 5 and 6.

### B.3  ABLATION EXPERIMENTS

To verify the necessity of each criterion in definition 2.1, we conduct experiments on the MNIST-CBCD dataset. Specifically, we conduct the similar experiments by removing ②, ③ and ④ in the objective function, and the corresponding results are shown in Fig. 4, 5, and  6 respectively.

|  |  | Raw | | | VAE | | | CCA-based | | | | MVAE | Ours | | |
|---|---|---|---|---|---|---|---|---|---|---|---|---|---|---|---|
|  |  | view1 | view2 | Concat | view1 | view2 | Concat | CCA | KCCA | DCCA | DCCAE |  | view1 | view2 | Concat |
| M | ACC | 11.7±0 | 26.8±2 | 12.2±0 | 35.4±2 | 38.5±4 | 56.2±3 | 46.9±3 | 48.2±2 | 40.2±3 | 47.4±4 | 50.4±7 | 62.0±3 | 62.1±3 | 62.4±2 |
|  | NMI | 0.3±0 | 18.2±2 | 0.5±0 | 25.0±3 | 30.6±3 | 50.2±2 | 54.2±3 | 52.9±1 | 51.3±2 | 51.9±3 | 41.9±3 | 54.8±1 | 56.4±1 | 57.0±1 |
| F | ACC | 15.1±1 | 36.8±2 | 28.7±2 | 42.5±3 | 45.6±3 | 52.0±3 | 48.6±3 | 46.2±2 | 42.5±3 | 46.4±3 | 51.8±4 | 56.1±3 | 54.9±3 | 56.3±2 |
|  | NMI | 3.9±1 | 32.3±2 | 24.5±1 | 41.1±2 | 45.1±3 | 55.1±3 | 55.1±3 | 56.7±2 | 52.8±3 | 54.0±2 | 54.1±2 | 58.2±2 | 58.5±2 | 59.4±2 |

Table 5: Comparison between existing multi-modal representation learning methods and ours on the clustering task. 'M' and 'F' indicate the MNIST-CBCD and FMNIST-CBCD datasets respectively.

|  |  | Raw | | | VAE | | | CCA-based | | | | MVAE | Ours | | |
|---|---|---|---|---|---|---|---|---|---|---|---|---|---|---|---|
|  |  | view1 | view2 | Concat | view1 | view2 | Concat | CCA | KCCA | DCCA | DCCAE |  | view1 | view2 | Concat |
| KNN | M | 77.8±.4 | 90.6±.5 | 77.9±.8 | 78.7±.9 | 89.3±.6 | 79.2±.6 | 83.0±.4 | 55.5±.9 | 52.9±.6 | 59.5±1.1 | 87.2±.8 | 88.1±.3 | 89.9±.6 | 91.2±.5 |
|  | F | 70.7±1.1 | 75.2±.3 | 67.8±1.1 | 74.8±.8 | 78.7±.6 | 74.9±.3 | 64.1±1.3 | 53.3±1.4 | 52.8±.6 | 56.8±1.4 | 73.2±.9 | 74.2±.7 | 78.1±.9 | 77.5±.9 |
| LSVM | M | 82.3±.7 | 90.4±.3 | 91.5±.3 | 85.6±.6 | 91.0±.7 | 91.1±.4 | 83.5±.7 | 61.8±.6 | 51.6±.8 | 64.9±.9 | 92.4±.4 | 89.2±.5 | 90.5±.6 | 92.2±.6 |
|  | F | 59.7±.2 | 78.1±1.5 | 79.6±.2 | 77.9±.8 | 81.3±.5 | 81.4±.6 | 65.1±.8 | 57.4±.6 | 47.1±.3 | 59.0±1.2 | 79.8±.2 | 79.4±1.0 | 81.9±.8 | 81.9±.8 |

Table 6: Comparison between existing multi-modal representation learning methods and ours on the classification task. 'KNN' and 'LSVM' indicate the K-Nearest Neighbor and linear SVM respectively.

As shown in Fig. 4, due to the removal of the intra-view exclusivity (②), there are shared information between $s^i$ and $e^i$, which is clearly validated by the reconstructed images (g) and (h). Similarly, as shown in Fig. 5, after removing the inter-view exclusivity (③), the performance of disentanglement becomes much worse. As shown in Fig. 6, after removing the condition ④, we can hardly disentangle the information from different views due to the significant difference between the shared representation $s^i$ and $s^j$.

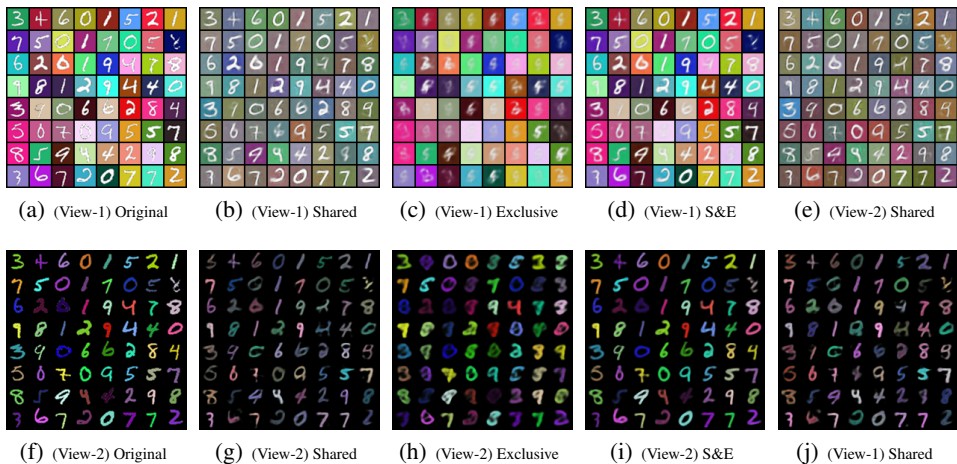

|  |  |  |  |  |
|---|---|---|---|---|
| (a) (View-1) Original | (b) (View-1) Shared | (c) (View-1) Exclusive | (d) (View-1) S&E | (e) (View-2) Shared |
| (f) (View-2) Original | (g) (View-2) Shared | (h) (View-2) Exclusive | (i) (View-2) S&E | (j) (View-1) Shared |

Figure 4: Visualization of reconstruction with shared and exclusive representations after removing the condition ② (intra-view exclusivity between $\mathbf{s}^i$ and $\mathbf{e}^i$). (Zoom in for best view).

### B.4 VERIFICATION: MAXIMIZING $\mathcal{I}(x; e) + \mathcal{I}(x; s)$ IS CONFLICT WITH MINIMIZING $\mathcal{I}(e; s)$

In Section 2.1, we analyze the reason why we do not maximize $I(x; s)$ and $I(x; e)$ to realize minimizing $\mathcal{I}(e; s)$. We provide qualitatively verification on the MNIST-CBCD dataset by only changing the way of maximizing $I(x; e, s)$. According to the experimental results in Fig. 7, we can find that both the shared representation (corresponding to Fig. 7(b) and (e)) and the exclusive representation (corresponding to Fig. 7(c) and (f)) contain almost the same information from the original views. This actually leads to poor disentanglement performance and also empirically validates the conflict discussed in Section 2.1.

### B.5 SUPPLEMENTAL VISUALIZATION RESULTS ON CELEBA

Furthermore, we conduct experiments on the face image-attribute dataset: CelebA Liu et al. (2015). The results are shown in Fig. 8. The reconstructed images with the shared representations from

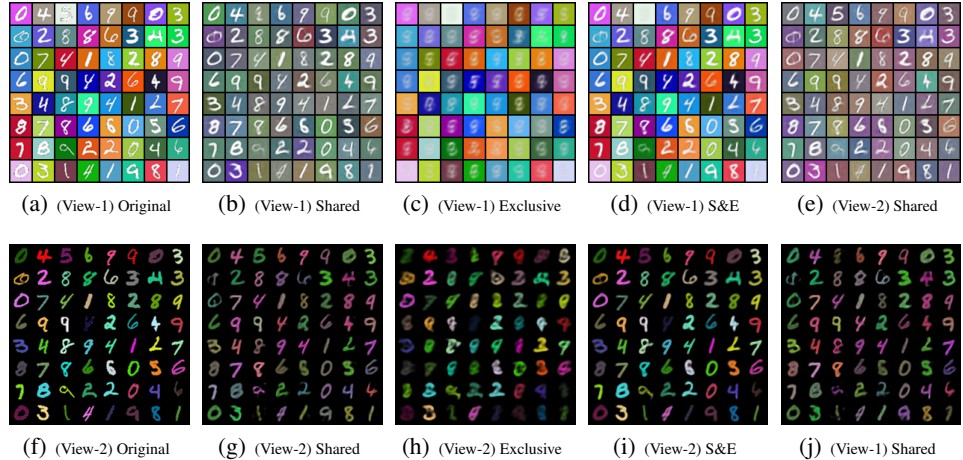

Figure 5: Visualization of reconstruction with shared and exclusive representations after removing the condition ③ (inter-view exclusivity between $\mathbf{e}^i$ and $\mathbf{e}^j$). (Zoom in for best view).

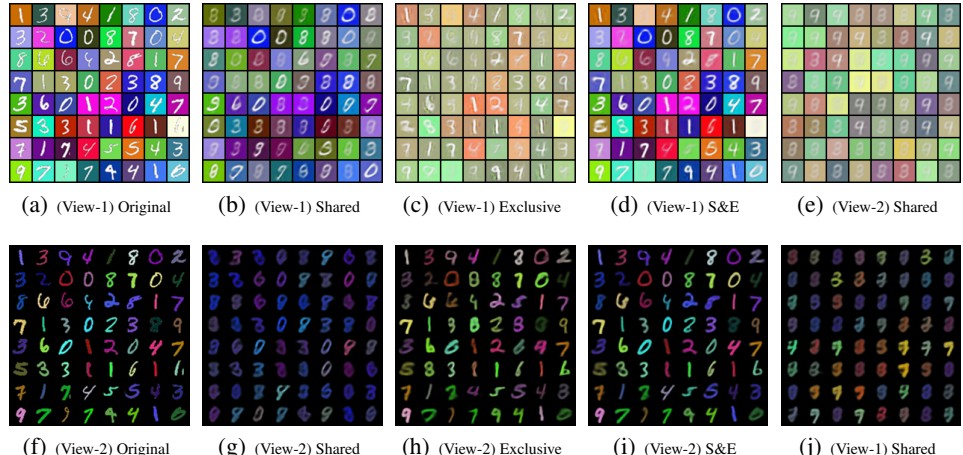

Figure 6: Visualization of reconstruction with shared and exclusive representations after removing the condition ④ (commonality between $\mathbf{s}^i$ and $\mathbf{s}^j$). (Zoom in for best view).

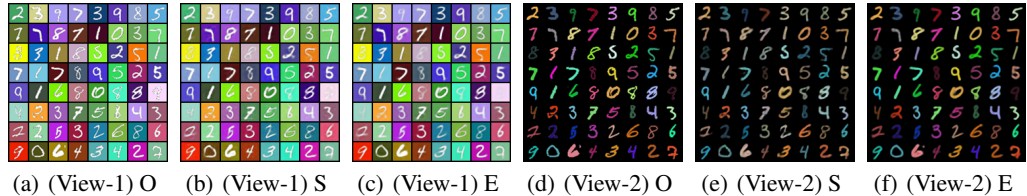

Figure 7: Visualization of reconstruction with shared and exclusive representations. 'O', 'S' and 'E' indicate original image, shared representation and exclusive representation respectively. 'View-1' and 'View-2' in the parentheses indicate the view where these representations come from. (Zoom in for best view).

different views ((b) and (e)) are relatively similar, and the results of (b) and (e) both reflect the same underlying attributes, including smile, hair color, hairstyle, gender (condition ④: commonality). The exclusive representation from the image view demonstrates that the reconstructed images correlate little to the attributes. For example, none of the reconstructed people have their mouths open and

their genders cannot be easily identified (condition ②: intra-view exclusivity). By combining the shared and exclusive representations, the original images can be accurately reconstructed (condition ①: completeness). Our model recovers the most critical information without emphasizing details because the current task is to obtain good representations for clustering/retrieval. By setting the goal to improve image reconstruction, we can use additional techniques (e.g., using more deeper networks - only 4 layers in our implementation, or using adversarial strategy). It is worth noting that there is a small difference from MNIST-CBCD: the information of view-2 (attributes) is actually contained in view-1 (images). Therefore, it is rather difficult to reconstruct face images using the exclusive representation from view-2 - the reconstructed images are almost all the same (condition ③: inter-view exclusivity). The visualization experiments on CelebA further verify that our disentangled representation can promisingly satisfy the four conditions in the definition 2.1.

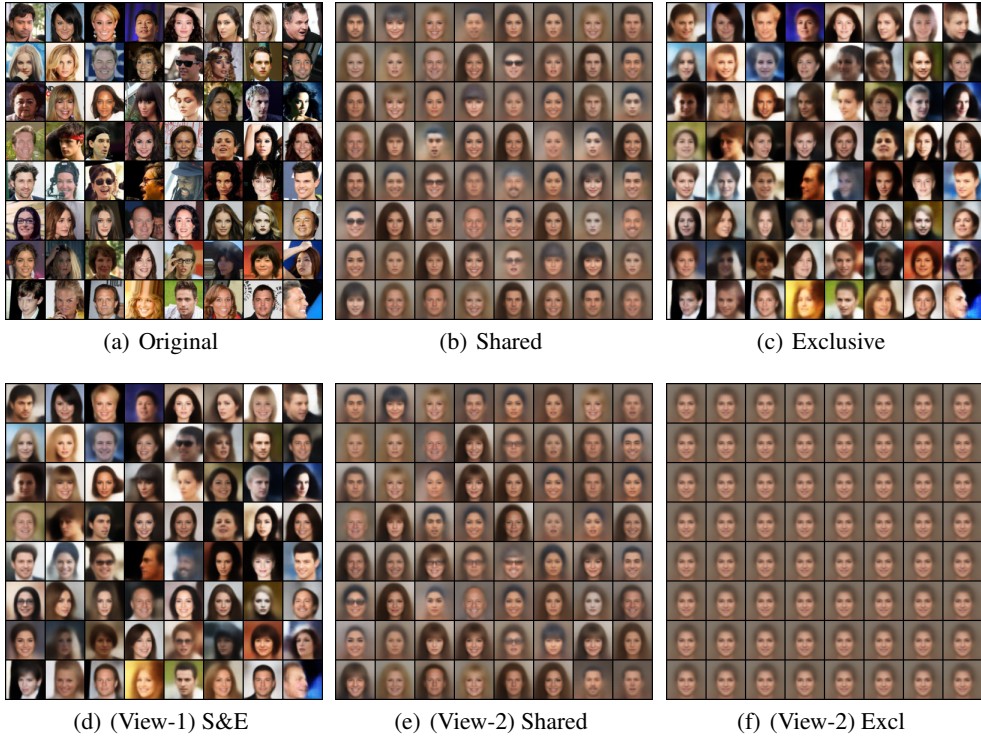

(a) Original       (b) Shared       (c) Exclusive

(d) (View-1) S&E       (e) (View-2) Shared       (f) (View-2) Excl

Figure 8: Visualization of image reconstruction with shared and exclusive representations. We use the decoder of the image view to reconstruct images by inputting the shared and exclusive representations, where 'Original' indicates the original images, 'Shared' and 'Exclusive' indicates the shared and exclusive representations, respectively. 'View-1 S&E' indicates the combination of shared and exclusive representations. Similarly, 'View-2 shared' and 'View-2 exclusive' indicate the shared and exclusive representations from View-2, respectively, which are used as inputs into the decoder. (Zoom in for best view).

## B.6 SUPPLEMENTAL RESULTS FOR ATTRIBUTE-SPECIFIC CROSS-MODAL RETRIEVAL AND EDITING

In this section, we validate the potential use of our multi-view disentangled representation in two real applications: attribute-specific retrieval and attribute-specific editing.

First, we verify our disentangled representation on the attribute-specific face retrieval task on the CelebA Liu et al. (2015) dataset. The details are as described in the text, and here we show some examples in Fig. 9(a).

Second, we demonstrate the potential use of our model in attribute-specific face editing by manipulating the shared representations. The shared representation from the image view allows us to

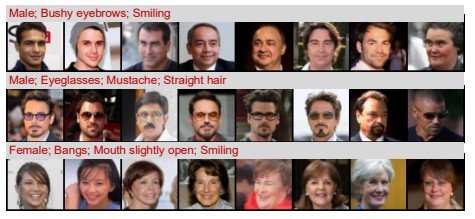 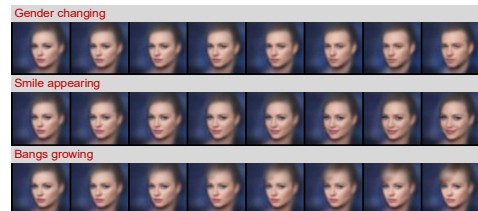

(a) Attribute-specific cross-modal retrieval (label-free for the gallery set).

(b) Attribute-specific editing.

Figure 9: Example results for face retrieval and face editing.

manipulate the specific properties of an image. The magnitude of the change can be controlled by interpolating between two shared representations. Specifically, to modify the visual properties of a person in an image, we can perform the following steps: (1) disentangling the shared ($s_{image}^o$) and exclusive ($e_{image}^o$) representations for a given image; (2) modifying the values in the attribute vector corresponding to the properties to be changed, and extracting the shared representation ($s_{attribute}^m$) from the modified vector; (3) replacing the original shared representation ($s_{image}^o$) with $s^{new}$, where $s^{new}$ is the linear interpolation between $s_{attribute}^m$ and $s_{image}^o$; (4) using $s^{new}$ and $e_{image}^o$ as the input to reconstruct the intended image. Representative experimental results are shown in Fig. 9(b).

We provide more results of attribute-specific editing for more attributes. The experimental results are shown in Fig. 10.

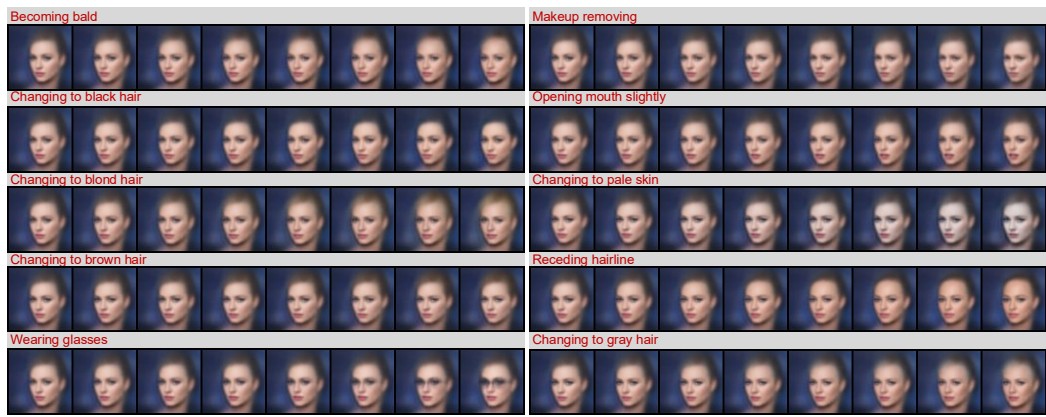

Figure 10: Example results for attribute-specific face editing.

## C  PROOFS

### C.1  PROOF OF $\mathcal{I}(x; e, s) = \int \int \int p(x, s, e) \log \frac{p(s, e|x)}{p(s, e)} ds de dx$

In order to obtain Eq. 3 in Section 2.1, first according to the chain rule for mutual information, we can get

$$\mathcal{I}(x; e, s) = \mathcal{I}(x; e) + \mathcal{I}(x; s|e), \tag{10}$$

where

$$\mathcal{I}(x; e) = \int \int p(x, e) \log \frac{p(x|e)}{p(x)} de dx$$
$$= \int \int p(x, e) \log p(x|e) de dx - \int \int p(x, e) \log p(x) de dx, \tag{11}$$

$$\mathcal{I}(x; s|e) = \int \int \int p(x, s, e) \log \frac{p(x|s, e)}{p(x|e)} ds de dx$$
$$= \int \int \int p(x, s, e) \log p(x|s, e) ds de dx - \int \int p(x, e) \log p(x|e) de dx. \tag{12}$$

Then $\mathcal{I}(x; e, s)$ can be formulated as

$$\mathcal{I}(x; e, s) = \int \int \int p(x, s, e) \log p(x|s, e) ds de dx - \int \int p(x, e) \log p(x) de dx$$
$$= \int \int \int p(x, s, e) \log p(x|s, e) ds de dx - \int \int \int p(x, s, e) \log p(x) ds de dx$$
$$= \int \int \int p(x, s, e) \log \frac{p(x|s, e)}{p(x)} ds de dx \tag{13}$$
$$= \int \int \int p(x, s, e) \log \frac{p(s, e|x)}{p(s, e)} ds de dx.$$

## C.2 PROOF OF $\mathcal{I}(x; e, s) = \mathcal{I}(x; e) + \mathcal{I}(x; s)$

Under Assumption 2.1, we can get $p(s, e) = p(s)p(e)$ and $p(s, e|x) = p(s|x)p(e|x)$. Substituting this into Eq. 13 yields Eq. 3 in Section 2.1,

$$\mathcal{I}(x; e, s) = \int \int \int p(x, e, s) \log \frac{p(e, s|x)}{p(e, s)} de ds dx$$
$$= \int \int \int p(x)p(e, s|x) \log \frac{p(e, s|x)}{p(e, s)} de ds dx$$
$$= \int \int \int p(x)p(e|x)p(s|x) \log \frac{p(e|x)p(s|x)}{p(e)p(s)} de ds dx \tag{14}$$
$$= \int \int \int p(x)p(e|x)p(s|x) \log \frac{p(e|x)p(s|x)}{p(e)p(s)} de ds dx$$
$$= \int \int p(x)p(e|x) \log \frac{p(e|x)}{p(e)} de dx + \int \int p(x)p(s|x) \log \frac{p(s|x)}{p(s)} de dx$$
$$= \mathcal{I}(x; e) + \mathcal{I}(x; s).$$

## C.3 PROOF OF $\mathcal{I}(x; e, s) = \mathcal{I}(x; r)$

First, based on Assumption 2.2, we can get

$$p(r) = \int p(r, x) dx = \int p(s, e, x) dx = p(s, e), \tag{15}$$

and

$$p(r|x) = \frac{p(r, x)}{p(x)} = \frac{p(s, e, x)}{p(x)} = p(s, e|x). \tag{16}$$

Accordingly, Eq. 6 in Section 2.1 can be derived as follows

$$\mathcal{I}(x; e, s) = \int \int \int p(x, e, s) \log \frac{p(e, s|x)}{p(e, s)} de ds dx$$
$$= \int \int \int p(x)p(e, s|x) \log \frac{p(e, s|x)}{p(e, s)} de ds dx \tag{17}$$
$$= \int \int p(x)p(r|x) \log \frac{p(r|x)}{p(r)} dr dx$$
$$= \mathcal{I}(x; r).$$

