# OpenReview forum: "Multi-View Disentangled Representation"
_ICLR.cc/2021/Conference — Reject_

### Official Review · AnonReviewer4 · 2020-10-28
**Clear definition of representation disentanglement**

**Rating:** 6
**Confidence:** 3

**Review:**

The goal of this paper is to define multi-view disentanglement in an unsupervised manner. The authors list four principal rules for representation disentanglement, including completeness of combining shared and specific representations, the exclusivity of two specific representations and between specific and shared representation, and commonality of two shared representation. The authors follow the above rules to design a VAE-based model and demonstrate favorable results on image clustering and classification tasks.

*Strengths*
1. Clear definition of representation disentanglement: as the prior works do not clearly define all constraints, which are ought to be valid for representation disentanglement, the prior models might not able to disentangle the representations into disjoint representations. This paper clarifies the four conditions and formally define the problem with four mutual-information terms. This sets the standard objective function for future works.

2. While the proposed model is based on VAE with modifications, the proposed model can perform favorably against existing works as presented in Table 1-4.


*Questions*
1. Extension to supervised representation disentanglement: I am wondering how to extend the proposed model to the supervised setting so that the semantic of the disentangled feature can be learned for further manipulation?

2. As beta-VAE is also designed for unsupervised disentanglement (although it is single-view), how beta-VAE performs in the tasks demonstrated in Table 1-4?

3. Generalization to more complicated data: I am wondering if the model is able to perform consistent improvement on a larger scale and more complicated dataset, as modeling mutual information in a higher dimension might be more difficult.

---

> ### Author Response · Authors · 2020-11-18
> **Response to AnonReviewer4**
>
> We thank the reviewer for valuable and positive comments. We try to reply each point as follows.
>
> **Q1**: Extension to supervised representation disentanglement: I am wondering how to extend the proposed model to the supervised setting so that the semantic of the disentangled feature can be learned for further manipulation?
>
> **R1**: An intuitive idea to extend the model to the supervised setting is to introduce semantic labels (attributes) as guidance. For example, considering there are two modalities (e.g., RGB image and depth image) and some possible attributes (as supervision information) for each modality, then the disentangled representations should be consistent with these attributes. Specifically, the shared representation should reflect the common attributes, and the private (exclusive) representation should reflect the private attributes as well. It is interesting and we will consider constructing a corresponding datasets and conducting experiments in the future.
>
> **Q2**: As beta-VAE is also designed for unsupervised disentanglement (although it is single-view), how beta-VAE performs in the tasks demonstrated in Table 1-4?
>
> **R2**: Since beta-VAE can only disentangle each view into possible semantic dimensions but cannot find shared and exclusive parts, for comparison, we can concatenate the disentangled features from each view, and we will conduct experiments within the following days.
>
> **Q3**: Generalization to more complicated data: I am wondering if the model is able to perform consistent improvement on a larger scale and more complicated dataset, as modeling mutual information in a higher dimension might be more difficult.
>
> **R3**: Thank you for your suggestion. We agree with the reviewer that modeling mutual information in a higher dimension is quite difficult, especially for such a total unsupervised setting. Even though it is difficult, we will explore the effect of the proposed model on more difficult data within the next days and will provide some results back.

---

### Official Review · AnonReviewer3 · 2020-10-28
**Good definition and implementation for multi-view disentangled representation.**

**Rating:** 5
**Confidence:** 5

**Review:**

For multi-view feature learning, this paper presents the definition as well as implementation for unsupervised multi-view disentanglement. Both consistency and complementarity are well investigated. Experiments on clustering and classification tasks validate the effectiveness.

Pros:
(1) The definition in 2.1 is quite straightforward, and the objective function in Eq. (1) is easy to understand.
(2) To measure the consistency and complementarity, this paper gives the solution based on mutual information.
(3) The presented experimental results look good.

Cons:
(1) I'm afraid that the consistency and complementarity have been investigated in multi-view clustering. [1,2] focus on affinity matrix based multi-view learning, and they already divided the affinity matrix of each view into the shared and the view-specific representations in an unsupervised manner. The difference lies in the format of input data and how to measure these properties. In view of this, the novelty of this paper is reduced.
(2) Please give more details about the optimization process.
(3) This paper focuses on multi-view learning. But the experiments only involve two views. How about the extension to more views. For example, you can conduct experiments on datasets with the number of views larger than 2 to validate the performance.
(4) In experiments, only s^{1], s^{2} and the concatenated representation are used for clustering or classification. Do you mean that the view-specific features are useless for the downstream tasks? Please give more explanation.

[1] Robust Multi-View Spectral Clustering via Low-Rank and Sparse Decomposition. AAAI 2014
[2] Exclusivity-Consistency Regularized Multi-view Subspace Clustering. CVPR 2017

---

> ### Author Response · Authors · 2020-11-18
> **Response to AnonReviewer3**
>
> Thanks for your valuable comments, and we hope the following responses can address your concerns.
>
> **Q1**: I'm afraid that the consistency and complementarity have been investigated in multi-view clustering. [1,2] focus on affinity matrix based multi-view learning, and they already divided the affinity matrix of each view into the shared and the view-specific representations in an unsupervised manner. The difference lies in the format of input data and how to measure these properties. In view of this, the novelty of this paper is reduced.
>
> **R1**: Thanks for the question. The two methods [1][2] mentioned are quite different from ours. One of the main novelties of our paper lies in the study what is a principled multi-view disentangled representation and provide a formal definition. The models [1][2] are not on this line.
>
> The motivation and function are quite different between these methods and ours. Both these two models do not explicitly decompose the shared and exclusive parts. The underlying assumption in [1] is the non-shared information is noise and thus they are modeled as error (E) to be minimized. While in [2], the authors do explicitly decompose the shared and exclusive parts, they only focus on maximizing the diversity between different views. [1][2] are not able to obtain disentangled representations at all. We will clarify this in the revision.
>
> **Q2**: Please give more details about the optimization process.
>
> **R2**: Thanks for the question. We will add detailed optimization process in the supplement. Actually, we mainly employed Adam optimizer to train the network and the objective function is simply the sum of several losses.
>
> **Q3**:This paper focuses on multi-view learning. But the experiments only involve two views. How about the extension to more views? For example, you can conduct experiments on datasets with the number of views larger than 2 to validate the performance.
>
> **R3**: Thanks for the comment. In the current version, we implement our algorithm for 2-view data. If consider data with more than 2 views, we can disentangle them in pairwise, i.e., disentangling each pair of views. We also admit that in this way it is not flexible for data with more than 2 views. How to elegantly extend our model for data with more than 2 views will be our future work.
>
> **Q4**: In experiments, only $s^{1}$, $s^{2}$ and the concatenated representation are used for clustering or classification. Do you mean that the view-specific features are useless for the downstream tasks? Please give more explanation.
>
> **R4**: To evaluate the disentanglement, we used $s^{1}$,$s^{2}$ (concatenated) to evaluate the shared part, while actually we also use the exclusive parts in our experiments, e.g., table 3 in section 4.1, face editing in section B.6. We also believe that it is also useful for downstream tasks in other scenarios.

---

### Official Review · AnonReviewer1 · 2020-10-28
**Clarifications on mutual information is needed**

**Rating:** 5
**Confidence:** 4

**Review:**

This paper proposes definition and conditions for unsupervised multi-view disentanglement providing general instructions for disentangling representations between different views. The authors also provide a novel objective function to explicitly disentangle the multi-view data into a shared part across different views and a (private) exclusive part within each view. I have the following comments on the paper.

major comments
1. In literature, there are several ways to estimate mutual information, such as lower bound of JS divergence (Hjelm 2019, Federici 2020), InfoNCE etc. Also there are other types of mutual information estimators that maximize similarities between two views. Could you provide any indication which of them work better than the other in you model?

2. I think the proposed model presented in the paper is inspired by the paper by Gonzalez-Garcia in 2018, where the authors didn't use the maximization and minimization of mutual information. Why a similar criteria based on mutual information performs better than the distance metric?

3. Maximizing mutual information is often considered as a difficult task and needs complicated sampling strategy (negative, hard negative etc). How do you handle those issues in your setting? Some details will be helpful for the community.

4. Without a bottleneck, i.e. assuming unbounded capacity, maximizing mutual information can be trivially solved by setting the underlying function to identity (Xu Ji et al. ICCV 2019). Have you tried any bottlenecking in your case? If not, how do you ensure that the mutual information maximization does not results in a degenerated solution. Do you think including a bootlenecking would improve the results?

5. The experimental evaluation only shows experimental comparison with some baseline approaches. I think it is also worth to compare the proposed methods with the prior works (for example, Gonzalez-Garcia et al. NeurIPS 2018).

Based on my current understanding and the above comments, I currently recommend the paper as "marginally below acceptance threshold". I would like to hear clarification on the proposed models and if satisfied would be happy to increase my recommendation.

minor comments
1. In ICLR 2020, there were few works that proposed to learn mutual information from diverse domains. I think it is worth to provide to have a discussion on them.
(i) M. Federici et al., Learning Robust Representations via Multi-View Information Bottleneck, ICLR, 2020.
(ii) M. Tschannen et al., On Mutual Information Maximization for Representation Learning, ICLR, 2020.
2. I think it worth providing some details on the implementation and architectures in the paper. I would also recommend to share the code.

---

> ### Author Response · Authors · 2020-11-18
> **Response to AnonReviewer1**
>
> Thanks for your valuable comments, and we hope the following responses can address your concerns.
>
> **Response to the major comments**
>
> **R1**: Mutual information estimation is still a difficult problem so there are different new ways as mentioned by the reviewer. In our work, we focus on the overall disentanglement where mutual information is only one component in our model. In practice, we provide discussion between using JS divergence (Hjelm 2019, Federici 2020) and KL-based MI, and we find they have different advantages as discussed in our paper (section A.2).
>
> **R2**: The main difference between ours and the paper by Gonzalez-Garcia in 2018 are as follows: (1) First, compared with our formal definition for multi-view disentanglement, [3] is a heuristic algorithm that explicitly decomposes the multi-view representation into shared and view-specific parts. There is not a formal definition provided for multi-view disentanglement, so as shown in Fig. 1, the disentanglement from their model may be unpromising (there is no guarantee for conditions 2 and 3); (2) Second, their method is especially designed for image-to-image translation tasks without disentangled representation guarantee and fall short in formally defining the relationships between different parts.
>
> **R3**: The strategy we use is similar to [a]. Specifically, we only use the representation of other samples in the same batch as negative samples. We will clarify this and release our code if the paper is accepted.
>
> [a] Hjelm R D, Fedorov A, Lavoie-Marchildon S, et al. Learning deep representations by mutual information estimation and maximization [C]//International Conference on Learning Representations. 2018.
>
> **R4**: We agree with the reviewer that in traditional unsupervised representation learning (with maximizing mutual information), if there is no constraint (e.g., bottlenecking) a degenerated solution may be obtained. Fortunately, in the proposed framework, there are four conditions acts as constraints that prevent our model from trivial solutions. Specifically, if setting the underlying function (neural networks) to identity, the loss will be very large because the four terms in objective (Eq. 1) will not be sufficiently optimized.
>
> **R5**: Since the method proposed by Gonzalez-Garcia et al. NeurIPS 2018 focuses more on image translation, its main novelty lies in the design of the network instead of representation learning. For example, it uses cross-domain autoencoders and GAN, which makes it not suitable for multi-modal data. Accordingly, it is quite difficult for us to conduct a fair experiment with this method.
>
> **Response to the minor comments**
>
> **R1**: Thanks for the suggestion, more discussions will be added.
>
> **R2**: In the appendix, we provide some implementation details. We will also release the code after acceptance.

---

### Official Review · AnonReviewer2 · 2020-10-28
**Interesting motivation, but the major claim needs solid proof.**

**Rating:** 5
**Confidence:** 3

**Review:**

This work aims to achieve disentangled representations for multi-view data. It explicitly summarizes the conditions for disentangled representations and proposed example solutions to achieve each of these criteria.

In my aspect, the three conditions for disentanglement is intuitional reasonable and can present insight for other works. However, I have the following concerns on this work:

1. the author presented the three conditions to be necessary conditions for disentanglement and argued them to be 'strict conditions'. However, I am afraid sufficient proof is needed to support these conditions to be necessary. In my opinion,  Information theory can be a candidate perspective to perform your proof, e.g. [1].

2. The proposed model achieves disentanglement on s and e with the product of the expert. So, will the design results in high computational cost at this phase?

3.  The compared baseline methods are not up-to-date.  This work specifically targets disentanglement, it is not fair to compare vanilla CCA methods which simply considers the shard information. VCCA-private [2] can be a good candidate to compare here.

4.  the work in [3] also specifically studies the multi-view disentanglement problem. Can you please discuss this method? It is also good to compare this method in the reconstruction visualization part.


[1] Gao, Shuyang, et al. "Auto-encoding total correlation explanation." The 22nd International Conference on Artificial Intelligence and Statistics. 2019.

[2].Wang, Weiran, et al. "Deep variational canonical correlation analysis." arXiv preprint arXiv:1610.03454 (2016).

[3]. Gonzalez-Garcia, Abel, Joost Van De Weijer, and Yoshua Bengio. "Image-to-image translation for cross-domain disentanglement." Advances in neural information processing systems. 2018.

---

> ### Author Response · Authors · 2020-11-18
> **Response to AnonReviewer2**
>
> Thanks for your valuable comments, and we hope the following responses can address your concerns.
>
> **R1**: Thank you for your constructive comments. In this paper, we define a multi-view disentangled representation and give an intuitive example (Fig. 1) to illustrate that when any condition of 1-4 is violated, the definition of disentanglement will not be satisfied. Note that, the four conditions are used to define the multi-view disentanglement, i.e., providing a formal definition of what is multi-view disentangled representation instead of a theorem or proposition.
>
> **R2**: The product of experts at this stage will not add significant additional computation. Generally, the computation complexity of PoE is $O(kn)$ where $k$ and $n$ are the feature dimensionality and the number of samples, respectively. The specific computation method can be found in the paper [a].
>
> [a] Wu M, Goodman N. Multimodal generative models for scalable weakly-supervised learning[C]//Advances in Neural Information Processing Systems. 2018: 5575-5585.
>
> **R3**: Thanks for the suggestion. Since there are quite a few multi-view disentanglement methods, in the current version we compared ours with CCA-based ones. We agree with the reviewer that VCCA-private is a good candidate comparison (although it is published on arxiv), and we will compare ours with VCCA-private during the rebuttal period.
>
> **R4**: First, compared with our formal definition for multi-view disentanglement, [3] is a heuristic algorithm that explicitly decomposes the multi-view representation into shared and view-specific parts. Second, the method [3] is especially designed for image-to-image translation tasks without a disentangled representation guarantee and falls short in formally defining the relationships between different parts.
>
> [3]. Gonzalez-Garcia, Abel, Joost Van De Weijer, and Yoshua Bengio. "Image-to-image translation for cross-domain disentanglement." Advances in neural information processing systems. 2018.

---

### Decision · Program_Chairs · 2021-01-07
**Final Decision**

**Decision:**

Reject

**Comment:**

This paper focuses on disentangled representation learning from multi-view data, which is an interesting and hot topic. However, there are several papers published in the last couple of years (especially in NeurIPS2020 and ECCV2020) solving very similar problems with closely related contributions to this paper. The contributions of this paper compared to all recent works in this space is unclear. Contributions and benefits of individual components in the method are not investigated. Although the method is designed for multi-view settings, the authors run experiments on simple settings with only two views. The experiments seem quite limited and do not show the method's capabilities. The rebuttal does not properly address the reviewers' concerns either.

The paper received four reviews with three recommending below acceptance threshold (rejection) and one above the acceptance threshold (although this one was the least confident scoring). Given all the above shortcomings and reviewer recommendations I do not recommend acceptance of the paper.